# Doxorubicin-Loaded Core–Shell UiO-66@SiO_2_ Metal–Organic Frameworks for Targeted Cellular Uptake and Cancer Treatment

**DOI:** 10.3390/pharmaceutics14071325

**Published:** 2022-06-23

**Authors:** Daria B. Trushina, Anastasiia Yu. Sapach, Olga A. Burachevskaia, Pavel V. Medvedev, Dmitry N. Khmelenin, Tatiana N. Borodina, Mikhail A. Soldatov, Vera V. Butova

**Affiliations:** 1Federal Research Center Crystallography and Photonics, Russian Academy of Sciences, 119991 Moscow, Russia; dirq@mail.ru (D.N.K.); borodina@crys.ras.ru (T.N.B.); 2Department of Biomedical Engineering, Sechenov First State Medical University, 119991 Moscow, Russia; anastsapach@gmail.com; 3Center for Neurobiology and Brain Restoration (CNBR), Skolkovo Institute of Science and Technology, 143025 Moscow, Russia; 4The Smart Materials Research Institute, Southern Federal University, 344090 Rostov-on-Don, Russia; oburachevskaya@sfedu.ru (O.A.B.); pavelmedvedev1994@gmail.com (P.V.M.); mikhailsoldatov@sfedu.ru (M.A.S.); butovav86@gmail.com (V.V.B.)

**Keywords:** nano-MOF, nanoparticles, UiO-66, MOF, silanization, silica shell, tumor targeting, folate receptors, chemotherapy, doxorubicin

## Abstract

Beneficial features of biocompatible high-capacity UiO-66 nanoparticles, mesoporous SiO_2_, and folate-conjugated pluronic F127 were combined to prepare the core–shell UiO-66@SiO_2_/F127-FA drug delivery carrier for targeted cellular uptake in cancer treatment. UiO-66 and UiO-66-NH_2_ nanoparticles with a narrow size and shape distribution were used to form a series of core–shell MOF@SiO_2_ structures. The duration of silanization was varied to change the thickness of the SiO_2_ shell, revealing a nonlinear dependence that was attributed to silicon penetration into the porous MOF structure. Doxorubicin encapsulation showed a similar final loading of 5.6 wt % for both uncoated and silica-coated particles, demonstrating the potential of the nanocomposite’s application in small molecule delivery. Silica coating improved the colloidal stability of the composites in a number of model physiological media, enabled grafting of target molecules to the surface, and prevented an uncontrolled release of their cargo, with the drawback of decreased overall porosity. Further modification of the particles with the conjugate of pluronic and folic acid was performed to improve the biocompatibility, prolong the blood circulation time, and target the encapsulated drug to the folate-expressing cancer cells. The final DOX-loaded UiO-66@SiO_2_/F127-FA nanoparticles were subjected to properties characterization and in vitro evaluation, including studies of internalization into cells and antitumor activity. Two cell lines were used: MCF-7 breast cancer cells, which have overexpressed folate receptors on the cell membranes, and RAW 264.7 macrophages without folate overexpression. These findings will provide a potential delivery system for DOX and increase the practical value of MOFs.

## 1. Introduction

Metal–organic frameworks (MOFs) are a class of porous crystalline polymers constructed by the self-assembly of metal–oxygen clusters and organic linkers, granting three-dimensional frameworks high surface area, exceptional periodically ordered and tunable pore size and topology, and easy access to functionalization [1,2]. Owing to the assembly of metal in different coordination forms with suitable organic ligands, MOFs have emerged as an enormous family with thousands of evolving members with tunable sizes and shapes of apertures that can be modified by design. Another unique advantage of MOFs lies in the fact that through in situ synthesis or post-synthetic modification, their structure can be functionalized with various chemical groups or metal ions, granting MOFs particular physical and chemical properties [1]. Initially, the main applications of MOFs were adsorption with influence in storage, separation, and purification. The list of applications includes acidic gas adsorption, toxic gas removal, gas storage, and water purification; the more advanced applications include different types of catalysis and doping processes, sensing, and biomedicine [1,3,4,5,6]. Porous MOFs have successfully been employed as drug delivery vehicles attributing to their enormous porosity, high surface area, and versatile framework compositions. Several studies have confirmed that MOFs exhibit exceptional biocompatibility and biodegradability [7,8].

Combining the high porosity and specific functionalities of MOFs (e.g., Lewis acid sites and the spatial and chemical control of the functional organic moieties) with the complementary properties of the other materials may enhance the properties of nanoparticles in the field of bioapplications. Multifunctional MOF composites can be made by growing or depositing a coating material around MOFs’ structure or mixing MOFs with other components. A series of composites are being developed, among them MOFs@metal or metal oxide, MOFs@quantum dots, MOFs@silica, MOFs@carbon, and MOFs@enzymes [7,9]. The surface modification of MOFs with a silica shell might offer several advantages such as improving their colloidal stability and ability to target specific cells or tissues; slowing down the biodegradation of MOFs, thus preventing an uncontrolled release of their cargo [10]. Synthesis of the MOF@silica composite allows for combining the ordered mesoporosity of SiO_2_ and microporosity of MOFs in a core–shell composition. MOF@SiO_2_ composites can be further functionalized with groups other than those typically used in MOFs, e.g., with a fluorophore and a cell-targeting peptide, which allows for their delivery to cancer cells and MRI [11]. The MOF@silica composite has silanol groups on their surface that can be employed for subsequent modification with molecules with siloxyl groups, which improves the targeting of certain types of cells. Thus, a small peptide sequence exhibiting a high binding affinity for many cancer cells was grafted to MOF@silica. Delivery of a platinum formulation using such a MOF@silica-peptide composite into colon cancer cells in vitro was almost two times more efficient compared to a nontargeted MOF@silica composite [12,13]. MOF@polymer composites are in high demand, as the polymer coatings impart additional functionality; in particular, they can strongly influence interactions with cells (for example, by PEGylation) [14]. Another example is the addition of pluronic F127 to the formulation, which is aimed at preventing the formation of a tightly bound hard corona layer, increasing the colloidal stability of the drug delivery system in the biological medium, and ensuring long-term circulation [15,16,17].

Zirconium-based MOFs of UiO-66 stand out from other nanoparticles due to the fact of their excellent thermal and chemical stabilities [8,18]. J. Winarta et al. gave a comprehensive review on the main properties and applications of UiO-66 MOFs including doped UiO-66 [4]. UiO-66 can be used without any chemical modifications such as for immobilizing enzymes to improve their storage stability and catalytic efficiency [6]. Zr possesses good biocompatible properties as well, due to the fact that humans consume approximately 3.5 mg per day. The in vivo toxicities of nanoscale MOFs, including UiO-66 and UiO-66-NH_2_, were studied [8]. For most of the MOFs, their respective cytotoxicity depended on the cell type and on the concentration. Interestingly, some MOFs, including UiO-66, showed little or no cytotoxicity, even at the highest dose of 200 µM [8]. None of the UiO MOFs exhibited substantial cytotoxicity at either incubation time, except for UiO-66-NH_2_; at a 200 µM dose and 24 h of incubation, this material showed moderate cytotoxicity to HepG2 cells. The good biocompatibility of UiO MOFs has been stated in [19,20,21].

Nanoparticles can be functionalized to impart new properties to MOF-based materials. In order to preserve the structure and size of the nanoparticles, post-synthetic functionalization of MOFs are of great interest. S. Nagarkar et al. functionalized UiO-66-NH_2_ nanoparticles with an azide group and showed fast and highly selective fluorescence turn-on response towards H_2_S under physiological conditions for the first time [22]. Low cytotoxicity and H_2_S detection in live cells, even in the presence of other relevant biomolecules, demonstrate the potential of MOFs towards monitoring H_2_S in the biological system. The imparting of highly hydrophobic properties to UiO-66-NH_2_ nanoparticles is shown by the microporous organic network surface coatings [23]. The obtained core–shell structures show excellent performance for adsorption of a model organic compound, toluene, in water, with potential use in detoxification.

UiO-66 shows a remarkable loading capacity for a wide range of compounds, along with other properties, and drug delivery by UiO-66 is one of the most attractive fields of application. It was evidenced that caffeine entrapping into UiO-66-NH_2_ reached payloads up to 22.4 wt % depending on the solvent choice, drug:MOF ratio, and ligand functionalization [24]. UiO-66 demonstrates an excellent ability to incorporate compounds with different molecular weights. The loading capacities of alendronate, 5-fluorouracil, calcein, ciprofloxacin, and cisplatin were 51, 27, 17, 84, and 12 wt %, respectively [14,25,26,27,28].

Until now, MOFs have been mainly proposed as anticancer delivery systems. In vitro and in vivo efficacy studies have focused on demonstrating the cytotoxic activity in cancer cell lines and in xenograft subcutaneous models. Photodynamic therapy in xenografted tumors in mice was successfully tested on photosensitizer-loaded UiO-66 nanoparticles, directly administered within the tumor [29]. X. Gao et al. demonstrated receptor-specific targeting of UiO-66-NH_2_ loaded with 5-fluorouracil and functionalized with folic acid in tumor-bearing mice [30]. Importantly, effectively targeted cancer treatment is realized by the sustained 5-fluorouracil release from UiO-66-NH_2_-folic acid-5-fluorouracil composites. After treatment with composite nanoparticles, the group showed inhibited tumor growth compared to the untreated group. UiO-66 was recently evaluated as a novel pulmonary drug delivery vehicle by investigating their aerodynamic properties upon aerosolization and degradability in extracellular- and intracellular-mimicking environments [31]. It was stated that UiO-66 exhibits high biocompatibility and low cytotoxicity in vitro and is well-tolerated in vivo in murine evaluations of orotracheally administered nanoparticles. Following pulmonary delivery, they remain intact, localized to the lungs before clearance over the course of seven days.

So far, it has been shown that along with Fe-based MOFs, UiO-66 can degrade rapidly in PBS pH 7.4 [24]. It was noted that the high water stability of UiO-66 dissapears very quickly in PBS, probably due to the strong affinity of Zr atoms for phosphate groups or the formation of zirconium oxide. Thus, the organic spacer or the number of complexing groups per spacer drastically influence the stability of the MOF in body fluid conditions. The most recent study of UiO-66 stability in buffers also suggests that the chemical nature of the buffer media played a decisive role in the stability, with a more pronounced leaching effect in the saline forms of these buffers [32]. The HEPES buffer was found to be the most benign, whereas MEM and PBS should be avoided at any concentration, as they were shown to degrade the UiO-66 framework rapidly. Low-concentration TRIS buffers are also recommended, although they offer a minimal buffer capacity to adjust pH. At the same time, there is evidence that UiO-66 could be stable in RPMI 1640 and DMEM cell culture media for 12 h [8,29]. Nevertheless, data collected to date are still very scarce, and more studies need to be performed. Nanoparticle degradation and aggregation as a result of structural changes in biological media will significantly alter the in vitro behavior (cellular uptake, cytotoxicity) as well as the in vivo fate (pharmacokinetics, toxicity, and biodistribution) of the nanoparticles [33]. As the in vivo studies show promising results in regard to the antitumor potential of MOFs, the issue of improving their stability in biological environments should be considered more carefully.

In this study, the objective was to fabricate a composite MOF@silica nanoparticle system for drug delivery. We combined the beneficial features of biocompatible high-capacity UiO-66 nanoparticles, mesoporous SiO_2_, and folate-conjugated pluronic F127 to prepare the core–shell UiO-66@SiO_2_/F127-FA drug delivery carrier, and then we carried out a series of characterization work. The colloidal stability of composite nanoparticles was studied in a number of model physiological media, including the most commonly used cell cultural media. Doxorubicin (DOX) was loaded onto nanoparticles to demonstrate its potential application in small molecule delivery. The final DOX-loaded UiO-66@SiO_2_/F127-FA nanoparticles were subjected to property characterization and in vitro evaluation including studies of internalization into cells and antitumor activity. Two cell lines were used: MCF-7 breast cancer cells, which have overexpressed folate receptors on the cell membranes, and RAW 264.7 macrophages without folate overexpression. These findings will provide a potential delivery system for DOX and increase the practical value of MOFs.

## 2. Materials and Methods

### 2.1. Chemicals

Poly(vinylpyrrolidone) (PVP, 10 kDa); zirconium tetrachloride (ZrCl4); benzoic acid (BA); 1,4-benzenedicarboxylic acid (BDC); 2-aminoterephthalic acid (BDC-NH2); *N*,*N*-dimethylformamide (DMF); 2-propanol; ammonium hydroxide (NH_3_·H_2_O, 28 wt %); tetraethyl orthosilicate (TEOS, 98%); poly(ethylene oxide)-block-poly(propylene oxide)-block-poly(ethylene oxide) (pluronic F127); folic acid (FA); *N*,*N*’-dicyclohexylcarbodiimide (DCC); dimethylaminopyridine (DMAP); dimethyl sulfoxide (DMSO); Hoechst 33258; Calcein AM; 3-(4,5-dimethylthiazol-2-yl)-2,5-diphenyltetrazolium bromide (MTT); dichloromethane; doxorubicin hydrochloride were purchased from Sigma-Aldrich (St. Louis, MO, USA). All chemicals were used as received without further purification.

Trypsin–EDTA solution (0.25% *v/v*); phosphate-buffered saline (PBS, pH 7.4), Versene solution, Dulbecco’s modified Eagle’s medium with phenol red (DMEM), Roswell Park Memorial Institute-1640 (RPMI1640) medium; fetal bovine serum (FBS), and MTT (thiazolyl blue tetrazolium bromide, 98%) were purchased from PAN-Biotech (Aidenbach, Germany). Deionized water from a three-stage Milli-Q Plus purification system was used in the experiments.

### 2.2. Synthesis of UiO-66 and UiO-66-NH_2_ Nanoparticles

UiO-66 and UiO-66-NH_2_ were obtained according to a previously reported technique [34,35]. In a typical synthesis, ZrCl_4_ was dissolved in DMF. After this, water was added to provide nuclei for UiO-66 formation. Then benzoic acid was added and the respective linker—terephthalic or amino-terephthalic acid. The molar ratio of the components ZrCl_4_:H_2_O:BA:Linker:DMF was 1:3:10:1:300. A clear reaction mixture was placed into a preheated oven at 120 °C for 24 h. After this period, the vessel was cooled down naturally. The precipitate was separated using centrifugation and washed two times with DMF and one time with 2-propanol. Obtained powders were dried at 60 °C for 12 h.

### 2.3. Synthesis of F127-FA Conjugate

The modification of pluronic F127 was performed by interactions of the hydroxyl groups of polyester and the carboxyl groups of folic acid (FA). Esterification reaction was conducted by the carbodiimide method using dicyclohexylcarbodiimide (DCC) as a condensing agent in the presence of catalytic amount of dimethylaminopyridine (DMAP) in dimethyl sulfoxide (DMSO) for 48 h at 4 °C. Namely, three solutions were prepared and mixed: 1.5 g F127 in 10 mL DMSO (1); 0.039 g FA, 0.011 g DMAP in 10 mL DMSO (2); 0.018 g DCC in 20 mL DMSO (3).

After completion of the reaction, unreacted FA was separated by washing twice with 5% sodium bicarbonate solution, and the resulting F127-FA conjugate was precipitated into hexane, filtered, and dried in a vacuum oven. Final purification of the reaction product from residues of FA, crosslinking agent, catalyst, and sodium bicarbonate was completed by dialysis (Servapor Dialysis Tubing, MWCO 10000, SERVAPOR, SERVA Electrophoresis GmbH, Heidelberg, Germany) from a chloroform–ethanol (1:9) mixture against water for 48 h, after which the solution was freeze-dried for 48 h.

### 2.4. Modification of the Surface of Nanoparticles with Silica Shells (Silanization of MOFs)

Generally, the MOFs@SiO_2_ composites are prepared by firstly modifying MOFs with a polymer to keep the MOF particles well dispersed and then coating them in a silica precursor solution. For this, 140 mg of UiO-66 was dispersed in 7 mL of poly(vinylpyrrolidone) solution (1 mg/mL) under ultrasound treatment and placed on a shaker at room temperature for overnight stirring. After centrifugation and washing with ethanol, PVP-coated UiO-66 MOFs were dispersed in 50 mL of ethanol. The aqueous ammonia solution (2 mL, 28 wt %) was added dropwise into the above UiO-66-PVP-containing ethanol under continuous vigorous stirring. After, 0.45 mL of TEOS was injected rapidly into the above solution and stirred for 0.5, 1, 2, 4, and 24 h. The obtained UiO-66@SiO_2_ composite nanoparticles were collected by centrifugation, washed three times with deionized water, and then dried in a fume hood. The same procedure was carried out with UiO-66-NH_2_ nanoparticles to synthesize the UiO-66-NH_2_@SiO_2_ composite.

### 2.5. Loading Nanoparticles with Doxorubicin and Modification with Folic Acid Conjugate

In a typical experiment, loading of DOX into UiO-66 and UiO-66@SiO_2_ composites was accomplished by mixing 1 mL of DOX solution (1 mg/mL) with 16 mg of MOFs. The mixture was placed on a shaker for 6 h under dark conditions. Free DOX was removed by centrifugation and washing with deionized water several times. The obtained DOX-loaded MOFs were stored at 4 °C in the dark. The amount of free DOX in the supernatant and washing solutions was determined by absorbance spectrum at an absorbance maximum of 481 nm using a linear calibration curve (Appendix A). The drug loading capacity (LC) was calculated according to the following formula:LC (wt %) = (weight of loaded DOX/weight of UiO-66 or UiO-66@SiO_2_) × 100%.(1)

To modify the particle surface with the F127-FA conjugate, UiO-66, UiO-66@SiO_2_, and DOX-loaded particles were mixed with the F127-FA solution (0.5 mg/mL) at room temperature for 1 h.

Samples were studied by transmission electron microscopy (TEM) in the bright-field (BF) mode, high-angle annular dark-field transmission scanning electron microscopy (HAADF-STEM), and energy-dispersive X-ray analysis using an electron microscope (Tecnai Osiris, FEI) operated at an accelerating voltage of 200 kV.

The samples in the powdered form were used for IR measurements. Measurements were performed on a Bruker Vertex 70 spectrometer equipped with a Bruker ATR-Platinum accessory and a liquid nitrogen-cooled MCT detector in ATR geometry (attenuated total reflection). Spectra were collected in the range 5000–30 cm^–1^ with a resolution of 1 cm^–1^ over 64 scans. Atmospheric compensation was used to eliminate CO_2_ and H_2_O’s influence. The reference background was recorded in an air atmosphere without a sample on the diamond crystal.

Powder X-ray diffraction data were collected on a Bruker Phaser D2 diffractometer using Cu Kα radiation (λ = 0.1541 nm).

UV–Visible (UV-vis) spectroscopy was performed with a PerkinElmer Lambda C650 spectrophotometer.

Dynamic light scattering (DLS) analysis was performed using a Zetasizer Nano ZS (Malvern, UK) at 25 °C.

### 2.6. Cell Cultures

Human breast adenocarcinoma MCF-7 cells and mouse macrophages RAW 264.7 were purchased from ATCC. Human breast adenocarcinoma MCF-7 cells were cultivated in DMEM supplemented with 10% FBS, 2 μM L-glutamine, 100 μg/mL streptomycin, and 100 U/mL penicillin in a 5% CO_2_ humidified atmosphere at 37 °C. The macrophage RAW 264.7 cell line was grown at 37 °C under 5% CO_2_ in RPMI 1640 culture medium containing 10% FBS and 1% antibiotics (i.e., 100 U/mL penicillin, 100 μg/mL streptomycin sulfate, and 0.25 U/mL L-glutamine). The cells were detached after treatment with a trypsin–EDTA solution (0.25% *v/v*), and the culture medium was replaced every 3–4 days.

#### 2.6.1. Flow Cytometry

For flow cytometry analysis, a BD FACSCalibur fluorescent-activated flow cytometer and BD CellQuest software were used. Cells were seeded in a 24-well plate (50,000 cells/well) followed by overnight incubation. Then, the culture medium was removed, and free DOX or DOX-loaded MOFs (50 μM DOX) were added to the cells and placed in the CO_2_ incubator (37 °C) or fridge (4 °C) for 0.5 and 2 h of incubation. After treatment, to remove noninternalized MOFs, the plates were washed three times with PBS (pH 7.4). Cells were harvested with a trypsin–Versene (ethylenediaminetetraacetic acid (EDTA)) solution, pelleted by centrifugation at 300 g for 5 min, resuspended in 200 μL of PBS (pH 7.4), and analyzed by flow cytometry with at least 10,000 cells being measured in each sample. The data are expressed as the median fluorescent intensity ± SD divided by the background intensity of the control (nontreated cells).

#### 2.6.2. Confocal Laser Scanning Microscopy

Cells were seeded on cover glasses (150,000 cells per glass) and incubated overnight. Then, the cells were incubated with the DOX-loaded MOFs (UiO-66@SiO_2_@F127-FA-DOX) resuspended in the culture medium (50 μM DOX) in a 5% CO_2_ humidified atmosphere at 37 °C. To visualize nuclei and cytoplasm, the cells were additionally stained with a Hoechst 33258 solution (50 μM, 10 min) and a Calcein AM solution (25 μM, 15 min), respectively. To remove noninternalized capsules, the plates were washed three times with PBS (pH 7.4). Then, the cells were mounted in a CC/Mount fluorophor protector and observed with a confocal laser scanning microscope (ZEISS LSM 880 Airyscan, Germany). The excitation wavelength values were 360, 488, and 543 nm for Hoechst 33258, Calcein AM, and DOX, respectively, while the fluorescence signals were collected at 380–460, 500–530, and 560–650 nm for Hoechst 33258, Calcein AM, and DOX, respectively. The images were processed in ZEN 2.3 pro blue edition software (Carl Zeiss, Oberkochen, Germany).

### 2.7. Cytotoxicity Study In Vitro

The cytotoxicity of the MOFs was studied by MTT assay. The cells were seeded in a 96-well plate (7500 cells/well) followed by incubation in CO_2_ incubator overnight. Free DOX or the DOX-loaded MOFs at various dilutions (0.5, 5, and 50 μM DOX) and blank MOFs (an amount equal to the DOX-loaded) were added to each well, and the cells were transferred to the CO_2_ incubator for 24 and 72 h. A monolayer culture (nontreated cells) was taken as a control (100%). After the treatment, the medium was replaced with an MTT solution in the culture medium (0.5 mg/mL) for 3 h. Then, the medium was replaced with DMSO (100 μL/well), and after complete dissolution of the formazan crystals, the absorbance was measured using a reader (Tecan Infinite^®^ 200 PRO) at 570 nm. The half-maximal inhibitory concentration (IC_50_) was determined as the drug concentration which resulted in 50% inhibition of cell growth.

### 2.8. Statistical Analysis

Statistical analysis of the experimental data was performed with GraphPad Prism, and the collected data were accepted as significantly different when *p* < 0.05. All experiments were carried out with at least three repetitions.

## 3. Results and Discussion

### 3.1. Synthesis and Characterization of UiO-66 and UiO-66-NH_2_

Synthesized samples of UiO-66 and UiO-66-NH_2_ were comprehensively characterized in our previous paper [35]. Both samples had a UiO-66-type structure with cubic symmetry, a Fm-3m space group (Appendix A). The morphology of the crystals was traced using TEM. Both samples were composed of octahedral crystals without significant aggregation Appendix A and Figure 1a–d). The average sizes of the particles in the UiO-66 and UiO-66-NH_2_ samples were estimated as 50–80 and 20–40 nm, respectively (Appendix A). The elemental analysis demonstrated a uniform distribution of carbon, zirconium, and oxygen over the UiO-66 particles (Figure 1e–h). As for the UiO-66-NH_2_ MOFs, the distribution of carbon, zirconium, oxygen, and nitrogen are almost uniform (Appendix A). UiO-66 are perfectly dispersed in water and formed a stable suspension of nanoparticles with a hydrodynamic size of 163 ± 53 nm (PDI 0.127) (Figure 2i). The size distribution by number, shown in Figure 2j, had a peak at 113 ± 38 nm. As expected, the average size calculated from the DLS measurements was slightly higher than from the TEM due to the effect of the dispersant on the hydrodynamic diameter and aggregation of UiO-66. However, the DLS numbers were close to the TEM results, whereas the DLS intensity presented a large difference with the TEM. Since the particle size distribution was not narrow, the presence of larger particles could increase light scattering, shifting the measured particle sizes towards larger values. 

The zeta potential of UiO-66 MOFs in water was +40 ± 1 mV. Since the particles are planned to be used for biological applications, they should remain stable in model environments such as saline solution and PBS. After placing the UiO-66 nanoparticles in 0.9% NaCl, MEM, DMEM, and RPMI cell culture media, their zeta potentials were −5 ± 3, −15 ± 1, −10 ± 1, and −21 ± 1 mV, respectively. A change in the sign of the zeta potential from positive to negative indicated that the particles adsorbed components of the dispersion medium—ions, amino acids, indicators, and others. This impairs their stability and promotes aggregation, which affects the size distribution: the average particle size exceeded 1 μm in all used cellular media (Appendix A). It should be noted that only in the RPMI did a part of the particles (approximately 7%) maintain their size at the nanoscale. A similar trend was observed for the UiO-66-NH_2_ particles, for which the zeta potential changed from +23 ± 2 mV in water to +2 ± 1, −6 ± 1, −5 ± 1, and 0 ± 2 mV in 0.9% NaCl, MEM, DMEM, and RPMI media, respectively. Their surface was modified with a porous silica shell to improve the colloidal stability of the UiO-66 and UiO-66-NH_2_ particles in biological media.

### 3.2. Synthesis of Core-Shell Structures UiO-66@SiO_2_

A series of core–shell nanoparticles were synthesized to study the effect of the silanization’s duration. For this purpose, the MOF particles were incubated in TEOS for 0.5 to 24 h after their activation with PVP. As expected, even the shortest TEOS incubation led to the deposition of silica on the nanoparticles (Figure 2a,f).

The HAADF-STEM images and elemental maps, shown in Figure 2, reveal an unexpected trend of SiO_2_ shell formation. An increase in TEOS incubation time did not result in a linear increase in shell thinness. The SiO_2_ shell on UiO-66@SiO_2_ particles incubated in TEOS for 2 and 4 h (Figure 2c,d,h,i) could hardly be distinguished. Elemental maps for MOFs treated with TEOS for 2 and 4 h showed almost uniform Zr and Si distributions. A similar result was obtained for UiO-66-NH_2_ nanoparticles coated with silica shells (Appendix A). The SiO_2_ shell’s thickness was determined by HAADF-STEM images analysis (Table 1).

At the same time, the results of the energy-dispersive X-ray spectroscopy during TEM (Figure 3) revealed that the amount of Si increased gradually with the increasing exposure time. This may mean that with rapid termination of the silicon deposition reaction (0.5–1 h), SiO_2_ has time to precipitate on the surface of nanoparticles, and with a further increase in the reaction duration (2–4 h), silicon penetrates into the highly porous structure of MOFs, impregnating them. Therefore, it was impossible to clearly distinguish a border between the MOF core and shell in the TEM images. The same process regarding silica formation inside the nanochannels of porous coordination polymers was previously described in [36,37]. A further increase in TEOS exposure up to 24 h led to the formation of a shell again on the surface of the silicon-impregnated MOFs.

Powder X-ray diffraction patterns (XRD) collected for intact UiO-66 nanoparticles and SiO_2_ nanoparticles (Sigma-Aldrich, nanopowder 5–15 nm, spherical, porous, 99.5%) were compared to those of the UiO-66@SiO_2_ and UiO-66-NH_2_@SiO_2_ samples treated with TEOS for different durations (Figure 4). The disappearance of the UiO-66 XRD pattern above 12° suggests a significant decrease in the crystallinity of UiO-66@SiO_2_ and UiO-66-NH_2_@SiO_2_ (a similar loss of crystallinity as a result of polymer modification was observed for another MOF [38]). However, the first two peaks at 7.4° (111) and 8.6° (002) could provide additional qualitative information about structural distortions. A shift in these peaks to higher degrees reveals the decrease of cell parameters in nanoparticles coated with a silica shell. Moreover, a higher intensity of these two peaks could be evidence of higher crystallinity in the UiO-66-NH_2_@SiO_2_ samples compared to UiO-66@SiO_2_.

Figure 5 represents nitrogen sorption isotherms of samples before and after coating. It can be observed that even after one hour, microporosity was significantly reduced. The specific surface areas of the samples were calculated according to BET modes. The values for samples UiO-66 and UiO-66-NH_2_ were estimated as 1354 and 1000 m^2^/g, respectively. The lower porosity of the UiO-66-NH_2_ sample could be assigned to amino groups located inside the pores, occupying part of the available volume. After coating for one hour, the specific surface area was reduced to 40 and 133 m^2^/g for UiO-66@SiO_2_ 1 h and UiO-66-NH_2_@SiO_2_ 1 h, respectively. Thus, a sample with amino groups preserved more available micropores than its analog UiO-66. We attribute this to the process of coating, which could lead, alternatively, to filling pores with silica or to the formation of the layer on the surface of the crystal. The former results in reduced microporosity, while the latter could add mesopores to microporous crystals. We suppose that UiO-66 during incubation with TEOS passes its molecules inside the pores, while amino groups of UiO-66-NH_2_ obstruct such a process. As a result, UiO-66-NH_2_@SiO_2_ 1 h sample exhibited a higher microporous volume. The total pore volume at P/P0 = 0.97 for the UiO-66@SiO_2_ 1 h sample was 55.5 mm^3^/g (Appendix A, Appendix A). The micropore volume for this sample was 9.0 mm^3^/g, according to t-plot calculations. Therefore, micropores contributed to only 16% of all porosity. The same calculations for the UiO-66-NH_2_@SiO_2_ 1 h sample resulted in 39% of the micropores. Pore size distribution was calculated according to the BJH model (Appendix A) and using the DFT approach (Appendix A). The main peaks at approximately 12 Å corresponded to octahedral pores of the UiO-66 framework. The UiO-66-NH_2_@SiO_2_ 1 h sample exhibited a slight peak position shift towards smaller pores due to the amino groups in octahedral pores. However, the total micropore volume of the UiO-66-NH_2_@SiO_2_ 1 h sample was higher. The UiO-66-NH_2_@SiO_2_ 1 h sample contained mesopores in the region of 10–18 nm, while the UiO-66@SiO_2_ 1 h sample comprised a silica shell with mesopores of 5–20 nm. The total volume of mesopores in the UiO-66@SiO_2_ 1 h sample was more significant.

The shape of the hysteresis loop of nonmodified UiO-66 and UiO-66-NH_2_ samples corresponded to type H1 in IUPAC notification. This could be attributed to the capillary condensation of nitrogen in the space between uniform spherical nanoparticles. According to the TEM images, both samples were composed of octahedral nanoparticles with a narrow size distribution. Their agglomerates could form such interparticle cavities. However, after 4 h of treatment with TEOS, both isotherms changed their shape. We did not observe steps in the low-pressure region. This indicates that micropores were unavailable for nitrogen. It should be noticed that all presented isotherms contained hysteresis loops in the region of relative pressures of 0.8–1. Nitrogen adsorption isotherms of the samples with silica shells contained hysteresis loops, which could be assigned to type H3. This indicates slit-like mesopores [39]. We attribute this to the formation of a mesoporous silica layer on the surface of microporous crystals.

The Fourier transform infrared (FTIR) spectra measured in an attenuated total reflectance (ATR) regime for intact and TEOS-treated UiO-66 and UiO-66-NH_2_ nanoparticles are shown in Figure 6 (FTIR spectra in a range from 4000 to 500 cm^−1^ are shown in Appendix A). The spectra of intact UiO-66 and UiO-66-NH_2_ nanoparticles were consistent with those previously reported and showed intense bands in the region of 1700–1200 cm^−1^ associated with carboxylate groups and phenyl ring deformations. An intense broadband at approximately 1050 cm^–1^ in the FTIR spectra of the TEOS-treated samples occurred due to the Si–O–Si asymmetric stretching vibration mode, while the peak appeared at 800 cm^−1^ attributed to the symmetric vibration mode of SiO_4_.

Our study reveals that surface silanization of UiO-66 and its derivative can be rather complex, and during the sol-gel process, silica can either internalize into the pores or form a shell. Authors publishing research on MOF@silica composites usually choose TEOS concentration and silanization duration arbitrarily starting from 2–3 h and ignore the study of the effect of the regime on the formation of the shell [11,12,13,40,41].

For another nanoparticle@silica composite, the shell synthesis conditions played an important role in the homogeneity and thickness of the SiO_2_ shell. There is a direct relationship between the duration of silanization and the thickness of the SiO_2_ layer for quantum dot@silica composites, which probably can be explained by the low porosity of quantum dots [42]. The shell thickness can be varied from 1 to 25 nm depending on the initial quantum dot size [43], and the typical duration for stable SiO_2_ shell condensation is no less than 20–24 h [42,44,45]. In the case of highly porous nanoparticles, the dependence may be nonlinear, as we demonstrated.

According to the EDX results, the Si/Zr ratio did not change within the error with an increasing duration of the silanization from 4 to 24 h for both UiO-66@SiO_2_ and UiO-66-NH_2_@SiO_2_ (Figure 3). Along with this, as can be seen from the TEM images, a distinct core–shell structure appeared after 24 h of silanization. To take full advantage of reproducible silica coating, including improved colloidal stability, ability to graft target molecules to the surface, and prevent an uncontrolled release of their cargo, further experiments were carried out with samples synthesized for 24 h.

### 3.3. Colloidal Stability Study in Model Biological Media

Treatment of bare UiO-66 MOFs with polyvinylpyrrolidone resulted in a change in the zeta potential value from a positive 39.7 ± 0.8 mV to a negative −14.9 ± 0.9 mV, which indicates successful surface grafting with PVP (Table 2). Since the zeta potential value was not very high, the formation of aggregates was noted—the DLS distribution of UiO-66/PVP was bimodal, and the PDI increased from 0.127 to 0.496 (Appendix A). After silanization, the zeta potential value in water increased up to −33.3 ± 0.4 mV, confirming the growing dispersion stability. Indeed, the size distribution of UiO-66@SiO_2_ was unimodal with a DLS number maximum of 247 ± 84 nm and a PDI of 0.172 (Appendix A). The shift in the distribution towards higher values was associated not only with the formed shell but, to a greater extent, with a change in the surface chemistry of nanoparticles and, most likely, with a thicker adsorption layer of ions.

When nanoparticles were placed into cell culture media, the absolute value of the zeta potential decreased but not as much as for uncoated UiO-66. UiO-66@SiO_2_ nanoparticles had the highest absolute value for the zeta potential in the PRMI medium, which correlated with the result for the uncoated UiO-66. Despite the similar values of zeta potentials, the DLS intensity distributions for UiO-66 and UiO-66@SiO_2_ dispersed in RPMI were completely different; only UiO-66@SiO_2_ particles had a peak in the nanoscale, which emphasizes the need to consider zeta potential values and size distributions together. The DLS intensity histograms of core–shell UiO-66@SiO_2_ reflects the fact that there were practically no aggregates in the samples dispersed in MEM, DMEM, and PRMI in contrast to uncoated UiO-66 (Appendix A). The PDIs for UiO-66@SiO_2_ core–shell particles in DMEM and RPMI culture media were estimated as 0.23 and 0.17, respectively, which is considered to be of moderate polydispersity according to ISO standard document 13321:1996 and ISO 22412:2008. The distribution peak for UiO-66@SiO_2_ dispersed in all culture media was at 310–330 nm (and the entire distribution was located in a range of less than 1 μm), while peaks for uncoated UiO-66 in MEM, DMEM, or RPMI were in the micron range. The broadening and shift in the distributions for UiO-66@SiO_2_ in cell culture media towards higher values were associated with the formation of a so-called protein corona. The results for nanoparticle dispersions in DMEM and RPMI media are of particular interest, as they culture cancer cells and macrophages for subsequent uptake and toxicity studies. Thus, the DLS revealed the good colloidal stability of UiO-66@SiO_2_ in cellular media.

### 3.4. Loading of MOFs with DOX

To study the DOX encapsulation efficiency of intact UiO-66 and UiO-66@SiO_2_, two MOF batches loaded with DOX were prepared. The amount of DOX was calculated according to equation (1) from the absorption spectrum using the plotted calibration curve (Appendix A). Firstly, for DOX impregnation, the particles were incubated in their solution for 1 h. The DOX loading capacity was 1.7 and 2.0 wt % for UiO-66 and UiO-66@SiO_2_ MOFs, respectively. Increasing the duration of incubation up to 6 h resulted in a more efficient loading: 6.2 and 5.8 wt % for UiO-66 and UiO-66@SiO_2_ MOFs, respectively. The high drug loading capacity of UiO-66 and UiO-66@SiO_2_ composite is probably attributed to the large surface area and porosity of the UiO-66 core and SiO_2_ shell. After loading, the particles were washed three times with deionized water to remove the excess drug. The final loading was 5.6 wt % for both uncoated and coated particles. Although the uncoated particles initially adsorbed more drug, it was easily released during the washings. This correlates with the fact that the porosity of uncoated particles was higher. Probably, the SiO_2_ shell retained the already adsorbed DOX from being washed out.

Further modification of the particles with the conjugate of pluronic and folic acid took 1 h, during which 3 and 12% of the encapsulated DOX was released from the UiO-66 and UiO-66@SiO_2_ particles, respectively (Appendix A). A sufficient decrease in the intensity of cargo release after silica coating was highlighted for nanoparticles from the MIL family [12] and Tb-based coordination polymers [13]. This result emphasizes the advantages of SiO_2_ coatings in preventing the undesired release of encapsulated molecules. It was shown that DOX loading did not influence the particle size distributions of both uncoated and coated MOFs (Appendix A). After modification with pluronic F127–folic acid conjugate, zeta potentials were estimated as −8 ± 1, −11 ± 1, −10 ± 1, and −17 ± 1 mV in water, MEM, DMEM, and RPMI media, respectively. The DLS number distributions for DOX-loaded UiO-66@F127-FA nanoparticles (Appendix A) indicated that the investigated colloids in MEM and DMEM cell cultural media had approximately 80% nanoparticles with a diameter of 199 ± 45 and 172 ± 31 nm, respectively, while the other 20% refers to aggregates. The size distribution for DOX-loaded UiO-66@F127-FA in RPMI was unimodal with a peak at 152 ± 100 nm and a tail up to 1 µm. DLS distribution for UiO-66@F127-FA in all model media correlated well with the distribution for the intact particles in water, which may indicate that the steric repulsion effect produced by the extended PEO chains of pluronic F127 prevented protein adsorption and also improved colloidal stability [15].

This surface modification had two goals. Firstly, F127 was aimed at improving the biocompatibility, prolonging the blood circulation time, and sensitizing drug-resistant cancer cells by providing enhanced drug transport across cellular barriers [46,47]. Folic acid, in turn, should target the encapsulated drug to the folate-expressing cancer cells, which we studied in vitro.

### 3.5. Evaluation of Cellular Uptake and Cytotoxicity Study In Vitro

Folate receptor (FR) is overexpressed in various cancer cells, while its expression in normal cells is restricted. FR-positive cells were chosen to study the effectiveness of the synthesized folate-conjugated block copolymer of F127 and to demonstrate its contribution to specific cell uptake. MCF-7 breast cancer cells, which have overexpressed FR in cell membranes [48], were incubated with nanoparticles in the normal conditions of the incubator at 37 and 4 °C. The internalization efficiency was estimated using fluorescence of doxorubicin-loaded into MOFs. In the cold, mammalian cells do not proliferate, and endocytosis can occur only through receptor-mediated mechanisms. The low fluorescence level in MCF-7 cells after cultivation with DOX-loaded UiO-66, DOX-loaded UiO-66-NH_2_, and free DOX at 4 °C supports this fact (Figure 7a). An increase in the level of fluorescence in cells cultured with MOFs having an F127-FA conjugate on the surface proves the activation of the folate-mediated internalization mechanism. Notably, the results for 0.5 and 2 h coincided, which indicates a quick completion of the receptor-mediated uptake [49]. In contrast, internalization at 37 °C may occur without recognition by specific receptors (often referred to as unspecific binding and unspecific uptake), possibly triggered by the nanosized object itself, as one can see in Figure 7b. An increasing uptake with an increase in the duration of cultivation is a classic feature of an unspecific process. The unspecific targeting ability of DOX-loaded MOFs was also evaluated using macrophage RAW 264.7 as the low-folate-receptor control. The MOFs’ uptake by macrophages in an incubator condition (Figure 7d) was practically the same as the cancer cells for all samples, but the MOFs’ uptake at 4 °C was negligible (Figure 7c).

Comparison of cellular uptake in folate-expressing and nonexpressing cells confirms the significant contribution of the folic acid conjugate to interactions between cells and nanoparticles.

The uncoated UiO-66 containers were rather leaky compared to UiO-66@SiO_2_ (Appendix A), and the presence of the released DOX in the cell culture medium may have contributed to the fluorescence intensity in Figure 7b,d. This, in addition to the specific endocytosis, is perhaps the reason why the fluorescence intensity in the case of the internalization of UiO-66@F127-FA nanoparticles was the highest in both cell lines.

Accumulation and localization of DOX-loaded UiO-66@F127-FA in cells were studied by confocal microscopy (Figure 8). As seen in Figure 8, for the case of the RAW 264.7 cells, the DOX-loaded UiO-66@F127-FA needed 2 h to go through the cytoplasm and partly accumulated at cell nuclei. However, some of the DOX-loaded MOFs continued presenting in the cytoplasm, confirming that DOX was released from the nanoparticles within the cells. In the case of specific targeting, the intensive fluorescence of the DOX-loaded UiO-66@F127-FA nanoparticles was observed at cell nuclei of MCF-7 cells. This could be explained by the differences in the DOX-loaded MOFs’ accumulation rates.

The steric repulsion effect produced by the extended PEO chains of pluronic F127 prevented protein adsorption and, thus, interactions with macrophages and cancer cells [15]. Despite this, DOX-loaded MOF nanoparticles effectively inhibit cell viability. Cytotoxicity values of free DOX and all samples of the DOX-loaded and blank MOFs were studied by MTT assay. As seen in Figure 9, no cytotoxic effect was revealed for the blank MOFs after 24 h; however, after 72 h, an insignificant effect on the viability of the cells was observed. After 24 and 72 h, the cytotoxicity levels of the DOX-loaded UiO-66 nanoparticles were higher than those of the DOX-loaded UiO-66@SiO_2_. These results could be explained by sustained DOX release from UiO-66@SiO_2_. As shown in [41], the drug release from Zn-based MOFs can be triggered by H^+^ cleavage of the coordination bond in an acidic environment, which is typical for lysosomes in a living cell. We can assume that the release of DOX from UiO-66 MOFs in cells occurred in this way.

Comparing the DOX-loaded UiO-66@SiO_2_ and UiO-66@SiO_2_@F127-FA toxicities, nanoparticles modified with SiO_2_ shell and F127-FA conjugate allowed more toxicity to be achieved for all DOX concentrations after 24 and 72 h due to the improved folate receptor-mediated cellular uptake. The maximum cytotoxicity was found for UiO-66@SiO_2_@F127-FA, which was in good agreement with the accumulation results (Figure 7). Therefore, it can be concluded that the cytotoxicity is in the function of the MOFs’ composition, accumulation efficiency, and DOX release profiles.

In general, the results demonstrate that the most specificity for targeting was achieved for the synthesized folate-conjugated block copolymer of F127. This effect was more pronounced for FR-positive cancer cells (MCF-7) compared to low-folate-receptor control- RAW 264.7 cells, even though the main function of macrophages is the uptake of foreign structures, specifically, various nanoparticles [50,51]. Normally, the expression of FR on the surface of most cells in the body is relatively low. The demand for folic acid increases during cell activation and proliferation. The expression of FR on the membrane of cancer cells is generally significantly higher than on normal cells [52]. Therefore, folate-modified targeted delivery systems have found application in cancer visualization and treatment as well as in the early detection of malignant neoplasms [53,54].

## 4. Conclusions

Our data demonstrate that silanization of highly porous MOFs may not occur in the same way as for low porous materials. Moreover, the concentration of reagents as well as the duration of the process play key roles in the silanization process. Particularly, 0.5 h silanization of UiO-66 nanoparticles showed formation of a thicker SiO_2_ shell (~9 nm), compared to those for UiO-66-NH_2_ (~5 nm). However, for longer silanization times (1 and 24 h), there were no significant differences (~5 and ~10 nm, respectively). The porosity of UiO-66 nanoparticles decreased dramatically (from 1354 to 40 m^2^/g) during the silica coating process. Interestingly, UiO-66-NH_2_ showed a moderate porosity decrease (from 1000 to 133 m^2^/g) after silanization. Core–shell UiO-66@SiO_2_ nanoparticles acquired colloidal stability in saline solution (~19 mV), MEM (~18 mV), DMEM (~−17 mV), and RPMI (~−21 mV) cell culture media, which is critical for its biomedical application. Together with improved stability, SiO_2_ shells provide the sustained release of the low molecular weight encapsulated doxorubicin compound. Further modification of the nanoparticles with the F127-FA conjugate was carried out to combine the improvement in their biocompatibility with active targeting of cancer cells.

Further, we evaluated UiO-66@SiO_2_/F127-FA nanoparticles loaded with doxorubicin with an efficiency of 5.6 wt % in vitro using folate-expressing MCF-7 breast cancer cells and RAW 264.7 macrophages without folate overexpression. The internalization studies show that the uptake of folic acid-coated MOFs occurs in a specific, receptor-mediated manner. Hence, F127-FA conjugate can be used for the active targeting of folate receptors, which are typically overexpressed in a variety of tumors. The MTT tests revealed the postponed onset of the toxic effect of the encapsulated DOX for MCF-7 cells. The two-component shell build-up from SiO_2_ and F127-FA provided a significant inhibitory effect of doxorubicin on cancer cells. We, therefore, highlight the potential application of core–shell UiO-66@SiO_2_@F127-FA nanoparticles in the field of drug delivery.

## Figures and Tables

**Figure 1 pharmaceutics-14-01325-f001:**
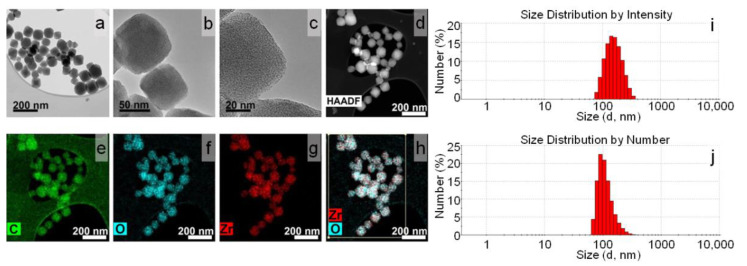
TEM/STEM images of intact UiO-66 MOFs and elemental mapping (**a**–**h**); DLS intensity (**i**); DLS number (**j**) distributions.

**Figure 2 pharmaceutics-14-01325-f002:**
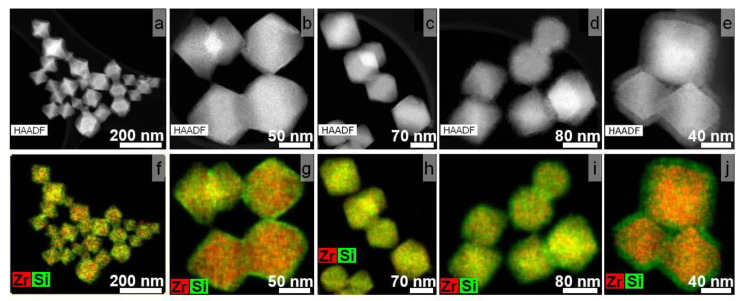
HAADF-STEM images and corresponding Zr-Si elemental maps for UiO-66@SiO_2_ samples obtained by incubation of PVP-grafted UiO-66 MOFs in TEOS for 0.5 (**a**,**f**); 1 (**b**,**g**); 2 (**c**,**h**); 4 (**d**,**i**); 24 h (**e**,**j**).

**Figure 3 pharmaceutics-14-01325-f003:**
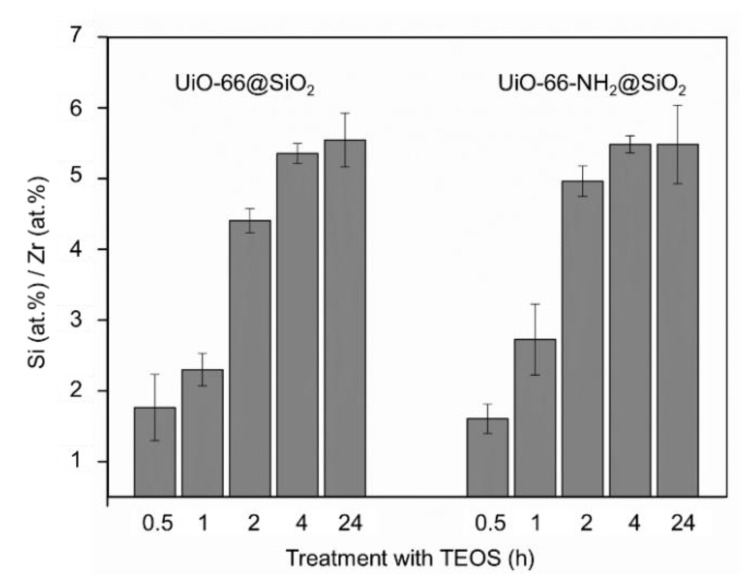
Si/Zr ratio in UiO-66@SiO_2_ and UiO-66-NH_2_@SiO_2_ MOFs treated with TEOS for different durations.

**Figure 4 pharmaceutics-14-01325-f004:**
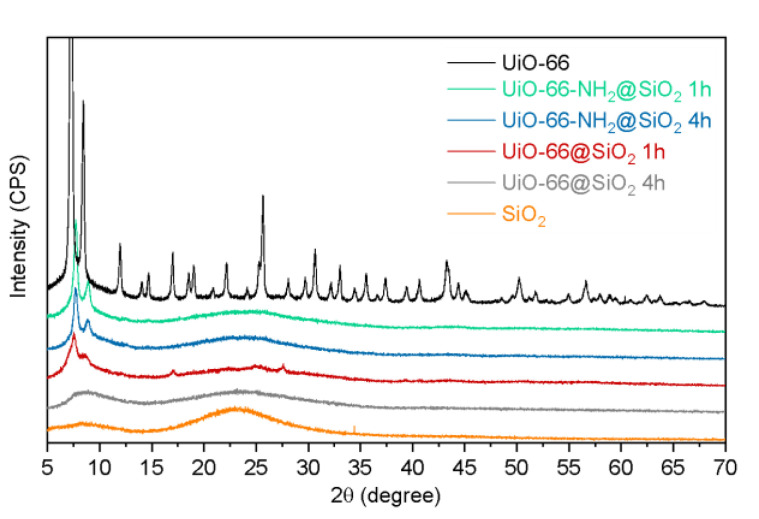
XRD patterns collected for intact UiO-66 nanoparticles, SiO_2_ nanoparticles, and TEOS-treated UiO-66@SiO_2_ and UiO-66-NH_2_@SiO_2_ samples.

**Figure 5 pharmaceutics-14-01325-f005:**
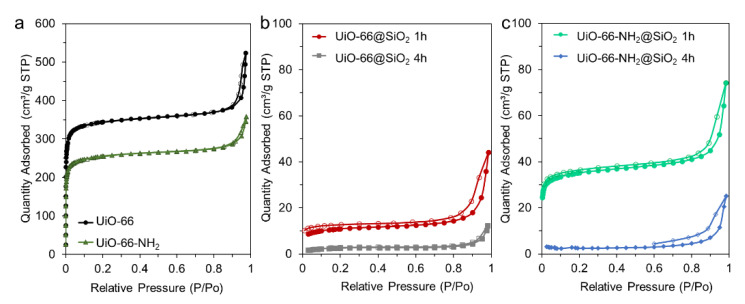
Nitrogen adsorption–desorption isotherms measured for UiO-66 and UiO-66-NH_2_ (**a**); UiO-66@SiO_2_ 1 h and UiO-66@SiO_2_ 4 h (**b**); UiO-66-NH_2_@SiO_2_ 1 h and UiO-66-NH_2_@SiO_2_ 4 h (**c**). Filled markers represent adsorption branches of isotherms, while empty ones designate desorption branches.

**Figure 6 pharmaceutics-14-01325-f006:**
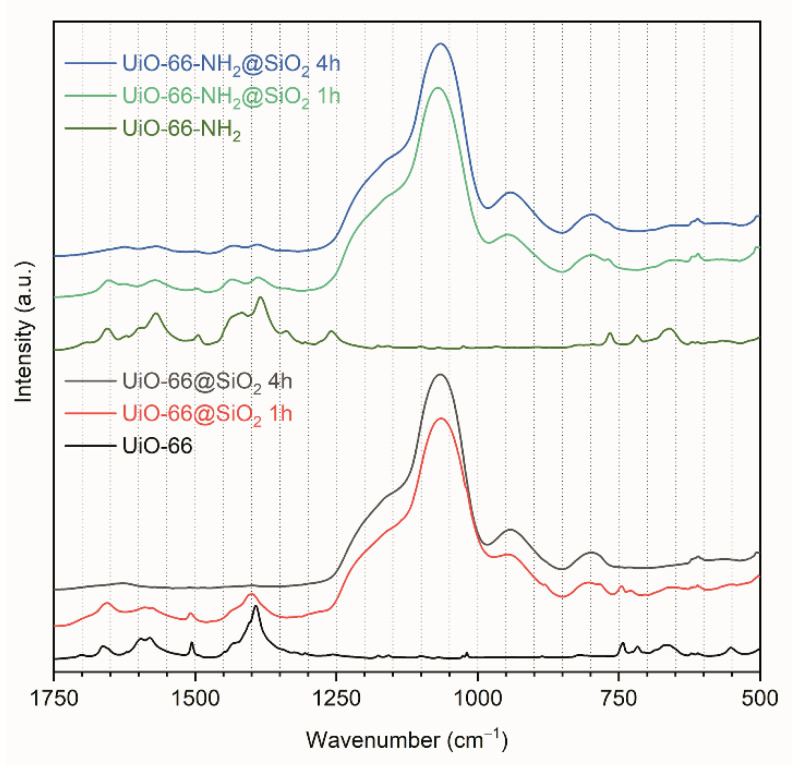
ATR-FTIR spectra collected for intact UiO-66 and UiO-66-NH2 nanoparticles compared to UiO-66@SiO_2_ and UiO-66-NH2@SiO_2_ MOFs treated with TEOS for different durations.

**Figure 7 pharmaceutics-14-01325-f007:**
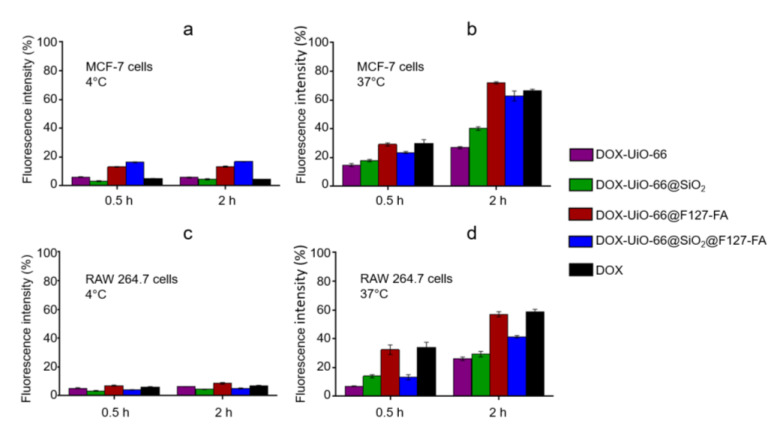
Uptake of the DOX-loaded MOFs and free DOX by MCF-7 cells at (**a**) at 4 °C and (**b**) 37 °C; and for RAW 264.7 w cells (**c**) at 4 °C; (**d**) at 37 °C after 0.5 and 2 h incubation durations.

**Figure 8 pharmaceutics-14-01325-f008:**
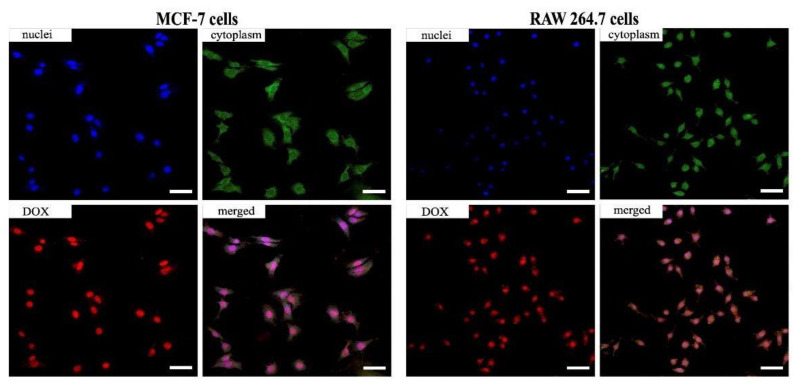
Intracellular distribution of the DOX-loaded MOF in MCF-7 and RAW 264.7 cells after 2 h. Cell nuclei are in blue (Hoechst 33258), cell cytoplasm in green (Calcein AM), and MOFs in red (DOX). Scale bar is 50 μm.

**Figure 9 pharmaceutics-14-01325-f009:**
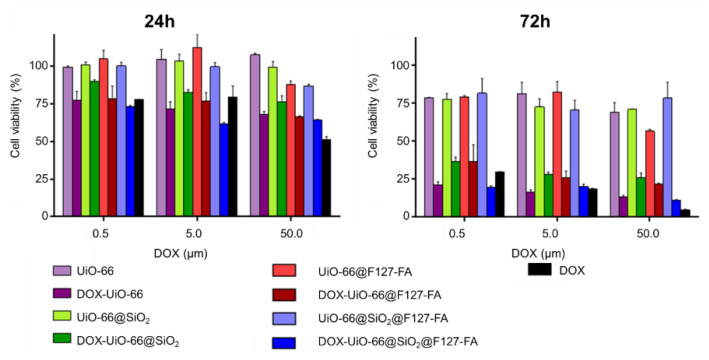
Cytotoxicity of the blank MOFs, DOX-loaded MOFs, and free DOX for MCF-7 cells after 24 and 72 h via MTT tests. A monolayer culture (nontreated cells) was used as a control (100%).

**Table 1 pharmaceutics-14-01325-t001:** Shell thickness in UiO-66@SiO_2_ and UiO-66-NH_2_@SiO_2_ MOFs depending on the duration of incubation in TEOS. n.a.—not available.

MOF Samples	The Thickness of SiO_2_ Shell after a Specific Duration of Silanization Process, nm				
	0.5 h	1 h	2 h	4 h	24 h
UiO-66@SiO_2_	8.5 ± 1.6	4.5 ± 0.9	n.a.	n.a.	9.9 ± 1.2
UiO-66-NH_2_@SiO_2_	4.5 ± 1.1	4.8 ± 1.1	n.a.	n.a.	9.9 ± 1.9

**Table 2 pharmaceutics-14-01325-t002:** Zeta potential of UiO-66, PVP-grafted UiO-66, and core–shell UiO-66@SiO_2_ in model media.

Sample	ξ of Nanoparticles in a Different Medium, mV				
	DI Water	Saline Solution	MEM	DMEM	RPMI
UiO-66	39.7 ± 0.8	−5.3 ± 3.1	−15.3 ± 0.9	−10.2 ± 0.6	−21.4 ± 0.9
UiO-66/PVP	−14.9 ± 0.9	−12.2 ± 1.3	−9.2 ± 0.3	−10.1 ± 0.7	−17.8 ± 0.9
UiO-66@SiO_2_ 24 h	−33.3 ± 0.4	+19.1 ± 1.2	−18.0 ± 0.5	−17.1 ± 0.9	−21.0 ± 1.5

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
