# Peer review of "Doxorubicin-Loaded Core–Shell UiO-66@SiO2 Metal–Organic Frameworks for Targeted Cellular Uptake and Cancer Treatment"

_pharmaceutics, 2022, doi:10.3390/pharmaceutics14071325_

Round 1

Reviewer 1 Report

The manuscript entitled “Doxorubicin-loaded core-shell UiO-66@SiO2 metal-organic frameworks for targeted cellular uptake and cancer treatment” is an interesting work, well done and well written. The authors explain the research clearly and simply. However, some aspects should be considered before accepting it for publication in pharmaceutics.            

Some aspects of the manuscript that the authors should improve are:

  1. The manuscript has minor grammatical, punctuation and writing errors, which should be carefully checked and corrected.

Results and discussion

  1. Page 8 Lines 316-317 …”The average sizes of particles in samples UiO-66 and UiO-66-NH2 were estimated as 50-80 nm and 20-40 nm, respectively (Fig. SI 1b)”..,

Why the particle size of the UiO-66-NH2 sample is smaller and with a narrower distribution than UiO-66 one? 

  1. What effect do the particle size and their distribution have on the proposed application? Analyze in greater depth and reference.
  2. Figure SI 1b requires revision and correction (the sample labels in the figure are wrong)
  3. Why XRD pattern of the UiO-66 sample in figure S1 1b is right-shifted? 
  4. The scales in the micrographs (Figures 1, 2, and 8) are not well observed, they must be improved
  5. Page 9 lines 338-343 “This impairs their stability and promotes aggregation, which affects the size distribution: the average particle size exceeds 1 μm in all used cellular media (Fig. SI 3 a-d). It should be noted that only in the RPMI, a part of the particles (about 7%) maintain their size in the nanoscale. A similar trend is observed for the UiO-66-NH2 particles, which zeta-potential changes from +23±2 mV in water to +2±1, -6±1, 342 -5±1, 0±2 mV in 0.9% NaCl, MEM, DMEM and RPMI media, respectively.” 

What is the mechanism of interaction of the particles with the medium? This is because in each case the particle size varies significantly. This should be explained in a more detailed and in-depth way.

  1. Figure 2 shows micrographs at different scales making it difficult to compare. Why not was used the same scale in all micrographs?
  2. Page 10 lines 365-368 “This may mean that with rapid termination of the silicon deposition reaction (0.5-1 h), SiO2 has time to precipitate on the surface of nanoparticles, and with a further increase in the reaction duration (2-4 h), silicon penetrates into the highly porous structure of MOFs, impregnating them.”

How is the size of these particles (at 2-4h) related to the original particles (before submerging them)? 

  1. Can the particle sizes at 0.5, 1 h, and 4 h be compared? If not are comparable, why? Please explain in detail.
  2. Page 10 lines 384-387 “The disappearance of the UiO-66 XRD pattern above 12° suggests a significant decrease of crystallinity UiO-66@SiO2 and UiO-66-NH2@SiO2.....”

            Does MOF lose crystallinity? How could it be possible? Please justify.

The thickness of the vitreous SiO2 coating that is deposited on the MOF surface could be enough to prevent the penetration of X-rays and as a consequence, the MOF structure is not detected.

  1. Page 11, lines 389-391 “qualitative information about structure distortions. A shift of these peaks to higher degrees reveals the decrease of cell parameters in nanoparticles coated with a silica shell.”

Why is there a decrease in cell parameters? 

  1. Were the cell parameters measured? 
  2. Why does the cell parameter decrease?
  3. A solid solution is promoted? hard data and references are required
  4. Page 11, lines 391-392 “Moreover, a higher intensity of these two peaks could be a hallmark of higher crystallinity of UiO-66-NH2@SiO2 samples compared to UiO-66@SiO2.”

What is the crystallinity of the samples before treatment with SiO2

  1. What is the thickness of the silica coating in both samples? 
  2. There is an effect of the amino group on the thickness of the silica coating? Why?
  3. Why the "hallmark of higher crystallinity of the UiO-66-NH2@SiO2 samples compared to UiO-66@SiO2"? What does it refer to?
  4. BET analysis. 

Since this analysis determines only mesoporosity, how do you determine the microporosity in the samples and the effect of the silica deposition on this microporosity? 

  1. What is the size and distribution of pores in the samples before and after silica deposition?
  2. Figure 6. In the FT-IR spectra of UiO-66-NH2@SiO2. Why not are observed the bands assigned to the amino group?

Author Response

Thank you for reviewing our work! Please find answers to all your questions in the file.

Reviewer 2 Report

In this paper, the authors have prepared a MOF and modified it with silica and then by a copolymer pf PEG/PPG. The nanomaterial is used for the delivery of DOX to cancer cell lines. The results are interesting and the paper is well presented. However, the manuscript lacks some parts that should be improved before consideration for publication.

1. The most significant issue is the application of these nanoparticles is well known and there are several reports on the modification of nanoparticles, especially a MOF, and this functionalization by polymers. For example, in a paper "Chem. 2017 Apr 13;2(4):561-78" the same MOF is modified by PEG and used as a carrier for drug delivery. The authors should discuss the novelty of their work and explain how their work differs from previous similar papers. Otherwise, I did not find any novelty in the paper and do not recommend it for publication. 

2. The authors have claimed that UiO-66 nanoparticles are biocompatible. They should cite a suitable reference to prove it.

3. The advantage of F127 instead of other similar polymers such as PEG has not been discussed. In the introduction section, the authors have cited a paper (the reference number 15 in the manuscript) and claimed that surface modification by F127 prevents the formation of a tightly bound hard corona layer and ensures long-term circulation. But in this paper, only the influence of different surfactants has been studied and the authors found that F127 is a better surfactant for the synthesis of their nanoparticles. The authors should reconsider this sentence (page 2, line 82).

4. The authors have claimed that "Doxorubicin encapsulation showed similar final loading of 5.6 wt.% for both uncoated and silica-coated particles". How coating has not affected the loading? Does it mean the loading of the drug into MOF and silica is the same? (while the coating by silica has not changed the loading et.% of the drug onto the nanoparticles).

5. How silica coating prevents the uncontrolled release of the drug? The authors should discuss if it is a delayed release or a controlled release that is responsive to stimuli.

6. again concerning the novelty of the work, the Modification of the nanoparticles by FA for delivery to cancer cells is not novel.

7. The use of two cell lines, one with overexpressed folate receptors on the cell membranes and one without that is an advantage. The authors can discuss the results of that more.

8. The English of the manuscript should be checked and revised. There are several typo and grammatical mistakes in the manuscript.

9. in N,N-Dimethylformamide, N should be italic.

10. in NH3*H2O, "*" should be replaced by ".".

11. In poly(ethylene oxide)-block-poly(propylene oxide)-block-poly(ethylene oxide), spaces between the characters should be removedl

12. In N,N’-dicyclohexylcarbodiimide, N should be italic.

13. There is no space between "F" and "127". in F127.

14. The sentences in the methods section should be in passive form. Page 5, line 192, the sentence "we have ..." should be rewritten.

15. Page 5, line 198, in 2-Propanol the letter "p" should be small.

16. Page 6 line 238, the equation should be numbered. It is easier for the readers to see that. Also for the authors, it is easier to cite the equation in the manuscript properly.

17. The conclusion section should be more quantitative.

Author Response

Thank you for reviewing our work! Please, find the answers to all questions below.

  1. The most significant issue is the application of these nanoparticles is well known and there are several reports on the modification of nanoparticles, especially a MOF, and this functionalization by polymers. For example, in a paper "Chem. 2017 Apr 13;2(4):561-78" the same MOF is modified by PEG and used as a carrier for drug delivery. The authors should discuss the novelty of their work and explain how their work differs from previous similar papers. Otherwise, I did not find any novelty in the paper and do not recommend it for publication. 

Authors answer: We agree with Reviewer concerning the broad application of the MOFs. These particles are very popular among the researchers due to their unique properties not limited to high surface area, exceptionally periodically ordered and tunable pore size and topology, and easy access to the functionalization. The novelty of the current paper based on the modification of the MOFs surface by silicification, ensures advantages of the both types of the particles for the final system. Moreover, we imparted targeted properties to the resulted MOFs and demonstrated their antitumor efficacy in vitro.

  1. The authors have claimed that UiO-66 nanoparticles are biocompatible. They should cite a suitable reference to prove it.

Authors answer: Thank you for the comment! We have cited articles reviewing the low toxicity of nanoparticles (please see Ref 6, 8). In the revised version of the manuscript, we have added fresh references establishing particle biocompatibility (Ref 19, 20, 21, Line 99).

  1. The advantage of F127 instead of other similar polymers such as PEG has not been discussed. In the introduction section, the authors have cited a paper (the reference number 15 in the manuscript) and claimed that surface modification by F127 prevents the formation of a tightly bound hard corona layer and ensures long-term circulation. But in this paper, only the influence of different surfactants has been studied and the authors found that F127 is a better surfactant for the synthesis of their nanoparticles. The authors should reconsider this sentence (page 2, line 82).

Authors answer: Thank you for the comment! We revised the mentioned sentence, clarified the role of F127, and add a few more references.

  1. The authors have claimed that "Doxorubicin encapsulation showed similar final loading of 5.6 wt.% for both uncoated and silica-coated particles". How coating has not affected the loading? Does it mean the loading of the drug into MOF and silica is the same? (while the coating by silica has not changed the loading et.% of the drug onto the nanoparticles).
  2. How silica coating prevents the uncontrolled release of the drug? The authors should discuss if it is a delayed release or a controlled release that is responsive to stimuli.

Authors answer: As we mentioned in the Section 3.4. Loading of MOFs with DOX, UiO-66 and UiO-66@SiO2 MOFs were loaded with DOX with loading efficiency of 6.2 and 5.8 wt.%, respectively.  After loading, the particles were washed three times with deionized water to remove the excess drug. The final loading after washings was 5.6 wt.% for both uncoated and coated particles. Although the uncoated particles initially adsorbed more drug, it was easily released during the washings. This correlates with the fact that the porosity of uncoated particles is higher. After, MOFs were incubated in F127-FA water solution for 1h, during which 3 and 12% of the encapsulated DOX were released from the UiO-66 and UiO-66@SiO2 particles, respectively (Fig. SI 8c). We assume, that SiO2 shell retains the already adsorbed DOX from being washed out. The same “barrier effect” of the SiO2  shell is noted by other authors in 10.1016/j.jre.2018.11.005.

  1. Again concerning the novelty of the work, the Modification of the nanoparticles by FA for delivery to cancer cells is not novel.

Authors answer: Modification of surface of the particles by FA is well known for imparting targeted properties to cancer cells. In the current paper, we used not only the FA-surface functionalization, but combined MOFs and Si particles and imparted targeted properties to the resulted particles by modification with Pluronic-FA conjugate demonstrated pronounce anticancer activity.

  1. The use of two cell lines, one with overexpressed folate receptors on the cell membranes and one without that is an advantage. The authors can discuss the results of that more.

Authors answer: We have included in discussion relevant information about folate receptor-mediated cellular uptake to make our manuscript more useful for a wide range of researchers. Find our detailed answer on the page 20 lines 650-660 (we also have added a new References 50 – 54):

In general, the results demonstrate that the most specificity for targeting was achieved for the synthesized folate-conjugated block copolymer of F127. This effect was more pronounced for FR-positive cancer cells (MCF-7) compared to low-folate-receptor control- RAW 264.7 cells, even though the main function of macrophages is the uptake of foreign structures, specifically, various nanoparticles [50,51]. Normally, the expression of FR on the surface of most cells in the body is relatively low. The demand for folic acid increases during cell activation and proliferation. The expression of FR on the membrane of cancer cells is generally significantly higher than on normal cells [52]. Therefore, folate-modified targeted delivery systems have found their application in cancer visualisation and treatment as well as in the early detection of malignant neoplasms [53,54].

  1. The English of the manuscript should be checked and revised. There are several typo and grammatical mistakes in the manuscript.

Authors answer: The language has been carefully checked and typos and errors have been corrected.

  1. in N,N-Dimethylformamide, N should be italic.
  2. in NH3*H2O, "*" should be replaced by ".".
  3. In poly(ethylene oxide)-block-poly(propylene oxide)-block-poly(ethylene oxide), spaces between the characters should be removed
  4. In N,N’-dicyclohexylcarbodiimide, N should be italic.
  5. There is no space between "F" and "127". in F127.
  6. The sentences in the methods section should be in passive form. Page 5, line 192, the sentence "we have ..." should be rewritten.
  7. Page 5, line 198, in 2-Propanol the letter "p" should be small.
  8. Page 6 line 238, the equation should be numbered. It is easier for the readers to see that. Also for the authors, it is easier to cite the equation in the manuscript properly.

Authors answer: Thank you for the comments! We have made the required corrections.

  1. The conclusion section should be more quantitative.

Authors answer: We added quantitative information to the conclusion section:

Our data demonstrate that silanization of highly porous MOFs may not occur in the same way as for low porous materials. Moreover, the concentration of reagents as well as the duration of the process plays a key role in silanization process. Particularly, 0.5 h silanization of UiO-66 nanoparticles showed formation of the thicker SiO2 shell (~9 nm), compared to those for UiO-66-NH2 (~5 nm). However, for longer silanization times (1 h and 24 h) there were not significant differences (~5 nm and ~10 nm, respectively). The porosity of UiO-66 nanoparticles decreases dramatically (from 1354 to 40 m2/g) during the silica coating process. Interestingly, UiO-66-NH2 showed moderate porosity decrease (from 1000 to 133 m2/g) after silanization. Core-shell UiO-66@SiO2 nanoparticles acquire colloidal stability in saline solution (~19 mV), MEM(~18 mV), DMEM (~-17 mV) and RPMI (~-21 mV) cell culture media, which is critical for its biomedical application. Together with improved stability, SiO2 shells provide the sustained release of the low molecular weight encapsulated doxorubicin compound. Further modification of the nanoparticles with the F127-FA conjugate was carried out to combine the improvement of their biocompatibility with active targeting of cancer cells.

Further, we evaluated UiO-66@SiO2/F127-FA nanoparticles loaded with doxorubicin with the efficiency of 5.6 wt.% in vitro using folate-expressing MCF-7 breast cancer cells and RAW 264.7 macrophages without folate overexpression. The internalization studies show that the uptake of folic acid-coated MOFs occurs in a specific, receptor-mediated manner. Hence, F127-FA conjugate can be used for the active targeting of folate receptors, which are typically overexpressed in a variety of tumors. The MTT test reveals the postponed onset of the toxic effect of the encapsulated DOX for MCF-7 cells. The two-component shell build up from SiO2 and F127-FA provides a significant inhibitory effect of doxorubicin on cancer cells. We, therefore, highlight the potential application of the core-shell UiO-66@SiO2@F127-FA nanoparticles in the field of drug delivery.

Reviewer 3 Report

The authors describe the synthesis, characterization and in vitro antitumoral activity of doxorubicin-loaded MOF particles. The manuscript is well-written and the results scientifically sound.

I noticed some details in the Materials and Methods that need to be fixed:

Line 174: “Poly(vinylpyrrolidone) (PVP), tetrachloride (ZrCl4)”: please state the average molecular mass of the PVP, and add “zirconium”.

Lines 201-207: the use of dichloromethane as a co-solvent is not stated as explicitly as that of DMSO. Please explain the choice of the solvent mixture.

Line 212: please state what kind of dialysis membrane was used.

Author Response

- Line 174: “Poly(vinylpyrrolidone) (PVP), tetrachloride (ZrCl4)”: please state the average molecular mass of the PVP, and add “zirconium”.

Authors answer: Information have been added.

- Lines 201-207: the use of dichloromethane as a co-solvent is not stated as explicitly as that of DMSO. Please explain the choice of the solvent mixture.

Authors answer: We would like to thank Reviewer for very important comment! Of course, the reaction was performed in DMSO solution. The corrections were made in the text. Please, see p.5.

- Line 212: please state what kind of dialysis membrane was used.

Authors answer: We used Servapor Dialysis Tubing: regenerated cellulose (MWCO 10000. The information was added to the paper. Please, see p. 6.

Reviewer 4 Report

The article describes a study of the preparation of UiO-66 @ Silica nanoparticle preparation, their loading with Doxorubicin and decoration with folates for targeted delivery applications.

The material characterization is thorough and well explained. The article clearly meets the quality criteria for publication in Pharmaceutics.

A few comments:

1) Considering that the authors demonstrate a loss of porosity with too long silicification, what is the reason of using samples silicified for 24h? Note that since the loading are comparable for UiO-66 and UiO-66@SiO2, blocked apparent porosity is not a big problem, and Dox can probably find a way toward the porosity in the conditions (even if N2 at 77K cannot for kinetics reasons...)

2) Considering the formation of silica in the pores, the authors may consider DOI: 10.1021/ja8030906, 10.1021/cm102610r and related references.

3) The manuscript is well written, but I could notice a few typos. Notably, some 2 in SiO2 are not subscript in the SI. Also, L.145, NEM, might be MEM. And maybe a few other small ones.

Author Response

Thank you for reviewing our work! Please, find the answers to all questions below.

- Considering that the authors demonstrate a loss of porosity with too long silicification, what is the reason of using samples silicified for 24h? Note that since the loading are comparable for UiO-66 and UiO-66@SiO2, blocked apparent porosity is not a big problem, and Dox can probably find a way toward the porosity in the conditions (even if N2 at 77K cannot for kinetics reasons...)

Authors answer: We performed the silicification process up to 24 hs investigating the silicon deposition reaction on the MOFs surface. The results gave us an important information, that silicon penetrates into the highly porous structure of MOFs, impregnating them. The incubation time increase leads to the gradual Si amount increase inside the MOFs according to results of energy-dispersive X-ray spectroscopy. We demonstrate that the 24-hs silicification allowed the Si shell formation on the surface of the silicon-impregnated MOFs. These results are very important for future application of the proposed system.

Moreover, the final comparable loading of UiO-66 and UiO-66@SiO2 with DOX was achieved due to the fact that initially more loaded uncoated particles released much more drug during the washing steps compare to Si-modified particles (Fig. S6 c). It means that the porosity of uncoated particles is higher, but the SiO2 shell retains the already adsorbed DOX from being washed out. The results are very important in the field of the development of the drug delivery systems with controlled release of the encapsulated components.

- Considering the formation of silica in the pores, the authors may consider DOI: 10.1021/ja8030906, 10.1021/cm102610r and related references.

Authors answer: Thank you very much for the recommendation. The information was added to the paper. Please, see p. 10.

- The manuscript is well written, but I could notice a few typos. Notably, some 2 in SiO2 are not subscript in the SI. Also, L.145, NEM, might be MEM. And maybe a few other small ones.

Authors answer: Thank you for the comment, the typo mentioned by the reviewer and a few more have been corrected.

Round 2

Reviewer 1 Report

The authors answered all questions in a good manner and the manuscript could be accepted in its present form.

Reviewer 2 Report

I recommend the publication of the manuscript. 

Reviewer 3 Report

The authors have properly amended the manuscript.